# Epidemiological, temporal, and geographic trends of leptospirosis in the United States, 2014–2020

Christine Atherstone[1]*, Renee Galloway[1], Ilana Schafer[1], Aileen Artus[1],
Melissa Marzan Rodriguez[2], Kyle Ryff[2,3], Abigail Medina Rivera[2], Sally Slavinski[4],
Marc Paladini[4], Sarah K. Kemble[5], Janet M. Berreman[6], Grayson Kallas[6], Rita Traxler[1],
Grishma Kharod[1], Marta Guerra[1], Robyn A. Stoddard[1], Hannah Moore[1], William A. Bower[1],
Maria E. Negron[1], Katherine DeBord[1]

1 Bacterial Special Pathogens Branch, Division of High-Consequence Pathogens and Pathology, National Center for Emerging and Zoonotic Infectious Diseases, CDC, Atlanta, Georgia, United States of America, 2 Puerto Rico Department of Health, San Juan, Puerto Rico, United States of America, 3 Field Assignment Branch, Division of State and Local Readiness, Office of Readiness and Response, CDC, Atlanta, Georgia, United States of America, 4 New York City Department of Health and Mental Hygiene, New York City, New York, United States of America, 5 Hawaii State Department of Health, Honolulu, Hawaii, United States of America, 6 Hawaii State Department of Health, Kauai, Hawaii, United States of America

* qdw2@cdc.gov

## Abstract

### Background

Leptospirosis, caused by pathogenic *Leptospira* spp., is one of the most widespread zoonotic diseases globally. In 2014, leptospirosis was reinstated as a nationally notifiable condition due to evidence of increasing incidence and public health importance. We describe the epidemiological, temporal, and geographic trends of leptospirosis in the United States since reinstatement.

### Methodology

Analysis included confirmed and probable leptospirosis cases from jurisdictions reporting ≥ 1 case between 2014–2020. Analyzed data included reportable case surveillance and voluntarily submitted supplemental data.

### Principal findings

Between 2014–2020, CDC received 1,053 case reports from 34 jurisdictions. The national incidence rate was 0.48 cases per 100,000 population. Since 2014, leptospirosis cases have been increasing, with an average annual gain of 13 cases ($R_2 = 0.69$). Cases increased in summer, peaking in early fall, corresponding with warmer weather and hurricane season. Among cases with outcome data, 85% (n = 606/709) were hospitalized and 10% (n = 74) died. Seventy-seven percent of cases (n = 623) reported

**Data availability statement:** National case surveillance data is available at: https://wonder.cdc.gov. Supplemental case report data is under the data ownership of each state/territory, and they cannot be shared publicly without state/territory permission. Please contact CDC at bspb@cdc.gov.

**Funding:** The author(s) received no specific funding for this work.

**Competing interests:** The authors have declared that no competing interests exist.

contact with animals or their bodily fluids while 71% (n = 578) of cases reported contact with freshwater or mud. More cases reported avocational activities (n = 413, [52%]) as the source of their animal or environmental exposure(s) than recreational or occupational activities (n = 203, [25%] vs n = 163, [20%], respectively). Only 13% of cases reported any international travel in the 30 days prior to symptom onset.

## Conclusions

An increasing number of leptospirosis cases in the U.S. are being reported, mostly from domestic sources of infection. Changing epidemiological trends away from occupational exposures to avocational or recreational activities highlights the need for interventions mitigating these exposure risks. A high percentage of cases were hospitalized and died emphasizing the need to educate healthcare providers, public health professionals, and the public about early identification and treatment for leptospirosis to improve patient outcomes.

## Author summary

Leptospirosis is a widespread zoonotic disease caused by pathogenic *Leptospira* bacterium, primarily affecting individuals in warmer climates. The World Health Organization estimates over a million cases and nearly 59,000 deaths annually worldwide. Individuals typically contract the disease through contact with urine from infected animals or urine-contaminated water, soil, or food. Symptoms can resemble other illnesses, complicating diagnosis, and while many cases are mild, 10–15% can lead to severe, life-threatening conditions.

This manuscript describes trends in leptospirosis cases reported to the Centers for Disease Control and Prevention from 2014 to 2020, following its reinstatement as a nationally notifiable disease. The findings indicate an increase in reported cases and a high percentage of cases requiring hospitalization and dying. Most cases were reported in Puerto Rico and Hawaii, with males aged 40–69 years being the most affected demographic. The study highlights a shift in risk factors, with recreational and avocational exposures becoming more common than occupational ones.

The report emphasizes the need for improved surveillance and public health interventions, particularly considering climate change projections of increased flooding and warmer temperatures, which may exacerbate the spread of leptospirosis. Enhanced awareness among healthcare providers and the public is crucial for early detection and treatment.

## Introduction

Leptospirosis, caused by pathogenic bacteria of the genus *Leptospira*, is considered one of the most widespread zoonotic diseases globally, with most cases occurring

in warmer climates [1,2]. The World Health Organization estimated at least one million clinical cases and 58,900 deaths annually worldwide, corresponding to 2.9 million disability-adjusted life years lost annually [2,3]. Infection in humans occurs through direct contact with the urine or other bodily fluids of infected animals or indirectly through contact with urine-contaminated water, soil, or food. Pathogenic leptospires enter the body through mucous membranes or abraded skin. Person-to-person transmission is rare.

Disease onset is typically two days to four weeks following exposure. Initial signs and symptoms can mimic other febrile illnesses, such as dengue or influenza [4,5], making diagnosis difficult [6]. Most infections are mild or asymptomatic [7]. However, approximately 10–15% of patients with clinical disease progress to severe, potentially fatal illness with multiorgan involvement that can include renal failure, liver failure, pulmonary hemorrhage, and meningitis [1,8]. Early initiation of antibiotics may help prevent more severe illness and decrease length of illness [9].

Historically, leptospirosis has been considered an occupational disease among those working outdoors or with animals [8]; however, groups at risk have expanded to include recreationally exposed populations [10,11] and individuals in areas experiencing significant weather such as hurricanes, flooding, or heavy rain [11–13]. Recreational exposures have included rafting, kayaking, and swimming in fresh water within the United States (U.S.) and in temperate and tropical regions of the world. Several leptospirosis outbreaks have been identified both in the U.S. and U.S. residents returning from recreational events abroad [14–16]. Numerous outbreaks after hurricanes, heavy rainfall, and flooding have been reported [17–21].

Between 1985 and 1994, annual cases of leptospirosis in the U.S. averaged 57, peaking in 1989 with 93 reported cases [22]. The annual incidence rate was 0.02-0.04 per 100,000 population during this time period [22]. Leptospirosis was removed from the list of nationally notifiable conditions in the U.S. in 1994. However, in some local, state, and territorial jurisdictions, leptospirosis remained a reportable condition. Between 1995 and 2010, some local, state, and territorial jurisdictions reported an increase in the number of cases [23,24]. Additionally, an outbreak occurred among participants of a triathlon in Illinois in 1998 [15] and in participants of an adventure race in Florida in 2005 [14]. Due to the growing evidence of its public health importance, leptospirosis was reinstated as a nationally notifiable condition in 2014 [25]. Since its reinstatement, this is the first surveillance summary describing the epidemiological, temporal, and geographic trends of leptospirosis in the U.S.

## Methods

### Data sources and collection

Non-disease specific data on leptospirosis cases are reported to the Centers for Disease Control and Prevention (CDC) electronically through the National Notifiable Diseases Surveillance System (NNDSS). While leptospirosis is a nationally notifiable condition, states or jurisdictions may opt not to include it on their list of reportable diseases or health conditions [26], and thereby not contribute to national case counts. States and jurisdictions can also voluntarily submit a supplemental leptospirosis case report form (CRF) to CDC, which contains additional information on occupation, animal, and environmental exposures; travel history; clinical signs and symptoms; treatment; outcome; and laboratory testing information. Supplemental case data is received by email, secure fax, or direct entry into the Data Collation and Integration for Public Health Event Response (DCIPHER) Program (Palantir Foundry, 2023). Leptospirosis case data submitted via NNDSS or supplemental CRF is compiled and stored in DCIPHER.

### Surveillance case definitions

The Council of State and Territorial Epidemiologists (CSTE) established a national case definition for leptospirosis in 2013 [24]. The 2013 CSTE leptospirosis definition includes clinical and laboratory criteria or an epidemiologic link; cases can be classified as confirmed or probable [24]. Confirmed cases meet at least one confirmatory laboratory criteria, which include

1) the isolation of *Leptospira* from a clinical specimen, 2) a fourfold or greater increase in *Leptospira* agglutination titer between acute- and convalescent-phase serum specimens studied at the same laboratory, 3) demonstration of *Leptospira* in tissue by indirect immunofluorescence, 4) a *Leptospira* agglutination titer of ≥ 800 by Microscopic Agglutination Test (MAT) in one or more specimens, or 5) detection of pathogenic *Leptospira* DNA (e.g., by PCR) from a clinical specimen. Probable cases are clinically compatible and either 1) meet at least one supportive laboratory criteria or 2) involvement in an exposure event (e.g., adventure race, triathlon, flooding) with associated laboratory-confirmed cases. Clinical criteria include a history of fever in the past two weeks and either 1) two of the following clinical findings: myalgia, headache, jaundice, conjunctival suffusion without purulent discharge, or rash (i.e., maculopapular or petechial) or 2) at least one of the following clinical findings: aseptic meningitis, gastrointestinal symptoms (e.g., abdominal pain, nausea, vomiting, diarrhea), pulmonary complications (e.g., cough, breathlessness, hemoptysis), cardiac arrhythmias or electrocardiogram abnormalities, renal insufficiency (e.g., anuria, oliguria), hemorrhage (e.g., intestinal, pulmonary, hematuria, hematemesis), or jaundice with acute renal failure. Supportive laboratory criteria include 1) *Leptospira* agglutination titer of ≥200 but <800 by MAT in one or more specimens, 2) demonstration of anti-*Leptospira* antibodies in a clinical specimen by indirect immunofluorescence, 3) demonstration of *Leptospira* in a clinical specimen by darkfield microscopy, or 4) detection of IgM antibodies against *Leptospira* in an acute phase serum specimen [24].

## Analyses

We analyzed NNDSS and supplemental CRF data from all confirmed and probable leptospirosis cases from all jurisdictions that reported at least one case between 2014–2020. Data from NNDSS included case counts, case classification, sex, age, race, and ethnicity. Data from supplemental CRFs included occupation, exposures (animal, environmental, travel), activity that led to exposure, reported clinical signs and symptoms, hospitalization events, antibiotic treatment received, outcome, and laboratory testing information (specimen collection date, test type, test result). Two categories were created to aggregate activities that led to exposure. Recreational activities included adventure race, triathlon, or mud run; biking or motorcycle riding; boating, kayaking, or rafting; camping or hiking; freshwater fishing; hunting; petting or touching animals at a farm, petting zoo, or other location; playing sports in yard or park; and swimming or bathing. The avocational category included drinking water; maintenance or house cleaning; pet or livestock ownership; and washing dishes or laundry. Occupational was a single response for an activity that led to exposure. State-specific and 10 year age-group incidence rates per 100,000 persons were calculated using 2020 U.S. Census Bureau state-specific total population data [27], decennial census of island areas (DECIA) [28], or the American community survey (ACS) [29]. Linear regression using the least squares method was used to assess the linear trend in the annual number of cases during 2014–2020. The Pearson product-moment correlation coefficient ($R_2$) was used to describe the proportion of variation explained by the model. Analyses were completed in DCIPHER (Palantir Foundry, 2023). This activity was reviewed by CDC and was conducted consistent with applicable federal law and CDC policy.

## Results

Between 2014 and 2020, 52 of 57 (91%) jurisdictions included leptospirosis on their list of mandated reportable conditions at some point. CDC received 1,053 case reports of leptospirosis in NNDSS from 65% (n = 34/52) of these jurisdictions. Of the 1,053 reported cases, 54% (n = 565) were classified as confirmed cases and 46% (n = 488), probable cases (Table 1). The overall incidence of leptospirosis in these jurisdictions was 0.48 cases per 100,000 population. Most of the leptospirosis cases were reported from Puerto Rico (n = 573, [54%]), Hawaii (n = 163, [15%]), California (n = 42, [4.0%]), and New York City (n = 38, [4.0%]). The highest incidence rates were reported from Puerto Rico (17.44 per 100,000 persons), Hawaii (11.51), Guam (6.50), US Virgin Islands (3.44), and Michigan (1.70) (Fig 1).

The highest annual case count was reported in 2017 (n = 193) followed by 2019 (n = 187) (Fig 2). Since 2014, the number of leptospirosis cases have been increasing with an average gain of 13 cases per year ($R^2 = 0.69$). Average weekly

**Table 1. Number and percentage of leptospirosis cases (n=1,053) by case classification, sex, age group, race and ethnicity – National Notifiable Diseases Surveillance System, United States, 2014-2020.**

| | Confirmed Cases, n=565 No. (%) | Probable Cases, n=488 No. (%) | Total Cases, n=1,053 No. (%) |
|---|---|---|---|
| **Sex** | | | |
| Female | 94 (17) | 98 (20) | 192 (18) |
| Male | 468 (83) | 390 (80) | 858 (82) |
| Unknown | 3 (0.5) | 0 (0) | 3 (0.3) |
| **Age group (yrs)** | | | |
| Minimum, maximum | 6, 93 | 0, 93 | 0, 93 |
| Median (Interquartile Range) | 48 (29) | 48 (29) | 48 (29) |
| 0-9 | 4 (0.7) | 11 (1.8) | 15 (1.2) |
| 10-19 | 46 (8.1) | 38 (7.8) | 84 (8.0) |
| 20-29 | 77 (14) | 55 (11) | 132 (13) |
| 30-39 | 78 (14) | 71 (15) | 149 (14) |
| 40-49 | 88 (16) | 70 (14) | 158 (15) |
| 50-59 | 120 (21) | 95 (20) | 215 (20) |
| 60-69 | 85 (15) | 85 (17) | 170 (16) |
| 70+ | 60 (11) | 49 (10) | 109 (10) |
| Unknown | 7 (1.2) | 14 (2.9) | 21 (2.0) |
| **Race** | | | |
| American Indian or Alaska Native | 1 (0.2) | 0 (0) | 1 (0.1) |
| Asian | 2 (0.4) | 5 (1.0) | 7 (0.7) |
| Black or African American | 12 (2.1) | 4 (0.8) | 16 (1.5) |
| Native Hawaiian or Other Pacific Islander | 1 (0.2) | 1 (0.2) | 2 (0.2) |
| White | 158 (28) | 122 (25) | 280 (27) |
| Other | 65 (12) | 48 (9.8) | 113 (11) |
| ≥2 races | 32 (5.7) | 20 (4.1) | 52 (4.9) |
| Unknown | 268 (51) | 281 (58) | 567 (54) |
| Missing | 8 (1.4) | 7 (1.4) | 15 (1.4) |
| **Ethnicity** | | | |
| Hispanic or Latino | 205 (36) | 197 (40) | 402 (38) |
| Not Hispanic or Latino | 166 (29) | 119 (24) | 285 (27) |
| Unknown | 190 (34) | 166 (34) | 356 (334) |
| Missing | 4 (0.7) | 6 (1.2) | 10 (0.9) |

case counts increased starting on epidemiological week 28 (mid-July) and peaked between weeks 40 and 43 (October) (Fig 3). More cases occurred in males (n=858, [81%]) than in females (n=192, [18%]). The age groups with the highest reported cases were 50–59 years (n=215, [20%], incidence rate: 0.70), 60–69 years (n=170, [16%], incidence rate: 0.61), and 40–49 years (n=158, [15%], incidence rate: 0.54) (Fig 4). More leptospirosis cases occurred in White persons (n=280, [27%; n=280/471, 59% among persons with known race]) and Hispanic or Latino persons (n=402, [38%; n=402/687, 59% among persons with known ethnicity]). However, only 19% (n=202/1,053) of cases had both race and ethnicity reported. More cases had missing or unknown race (n=582, [55%]) than missing or unknown ethnicity (n=366, [35%]).

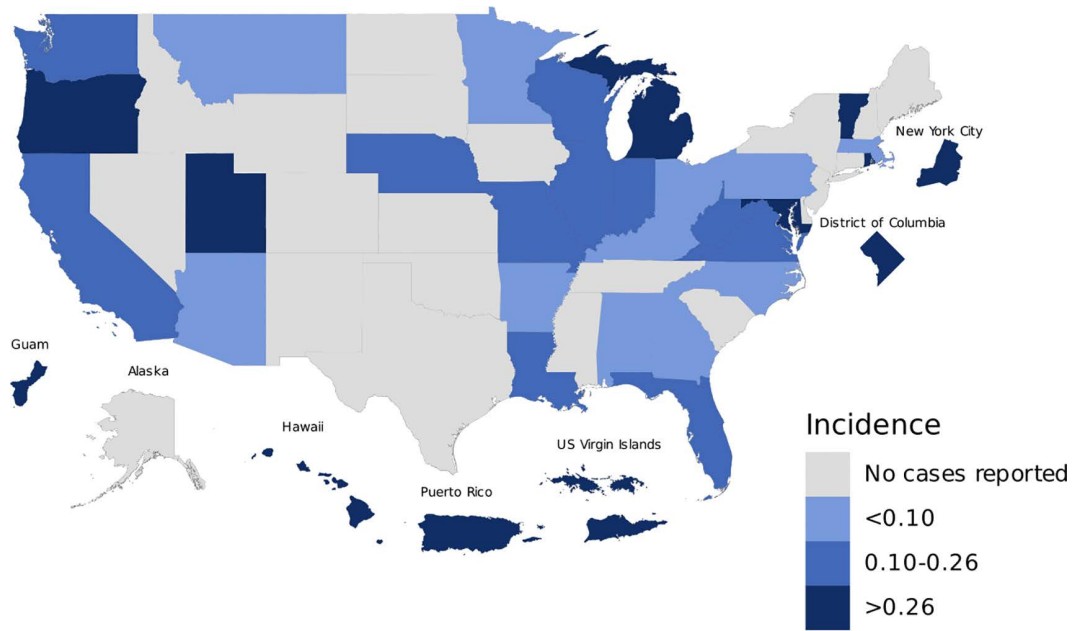

**Fig 1. Cumulative incidence rate (per 100,000 persons) of leptospirosis, by jurisdiction – National Notifiable Diseases Surveillance System, United States, 2014-2020\*.** \*Jurisdictions where leptospirosis was not locally reportable during a subset of the analytic period: American Samoa (2014-2015), AR (2014-2019), ID (2018-2020), IA (2015-2020), KS (2014-2017, 2019), KY (2014-2016), MA (2017, 2020), NH (2015-2016), NY (2015-2020), ND (2014-2019), SD (2014-2015), VA (2015-2016); Not reportable during analytic period: CO (2014-2020), CT (2014-2020), MS (2014-2020), TN (2014-2020), TX (2014-2020). Map Base Layer: https://epsg.io/4269.

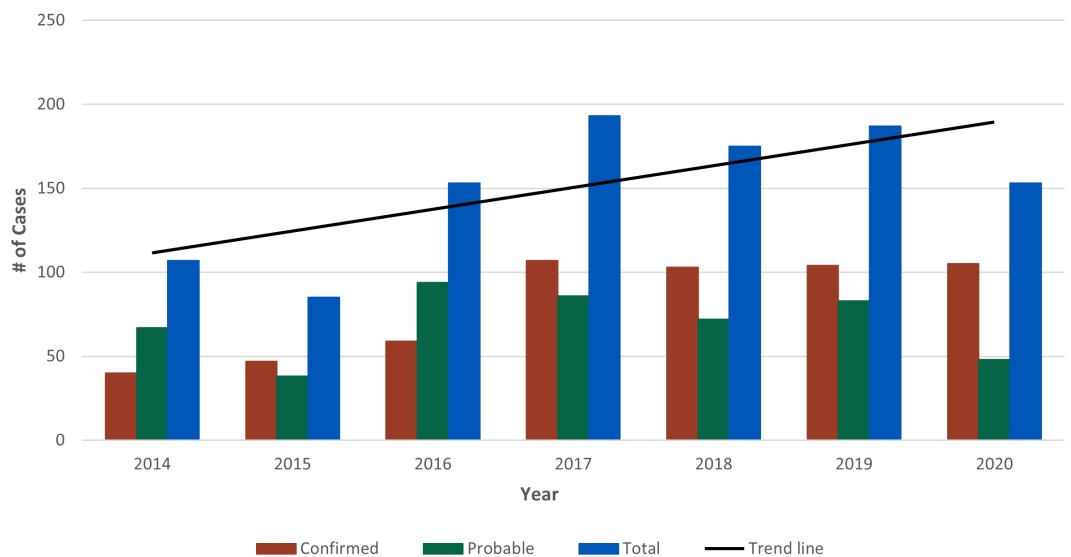

**Fig 2. Total number of leptospirosis cases by year - National Notifiable Diseases Surveillance System, United States, 2014-2020\*.** \*The COVID-19 pandemic may have affected disease detection and reporting in 2020.

Among the 1,053 cases reported in NNDSS, 77% (n = 809) had a supplemental CRF voluntarily submitted to CDC (Table 2). When comparing case classification reported on supplemental CRFs, which contained laboratory test results, against CSTE case classification criteria, 96% (n = 461/481) of confirmed and 57% (n = 187/328) of probable cases met

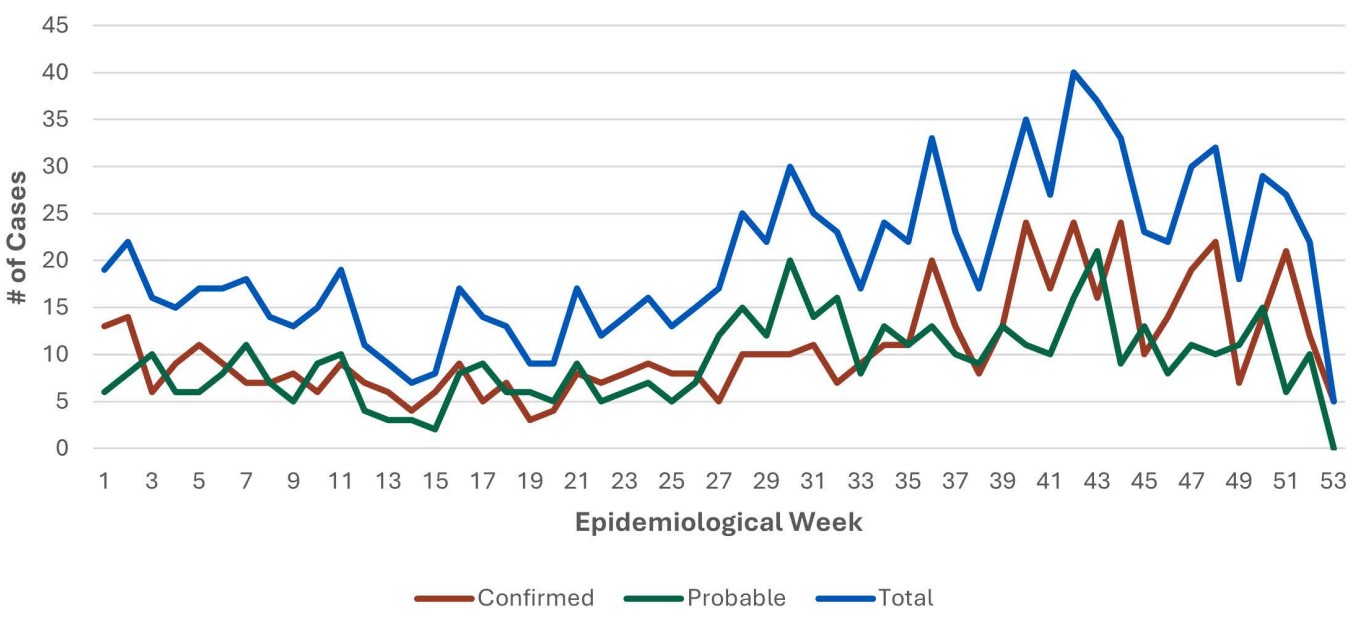

**Fig 3. Total number of leptospirosis cases by epidemiological week - National Notifiable Diseases Surveillance System, United States, 2014-2020.**

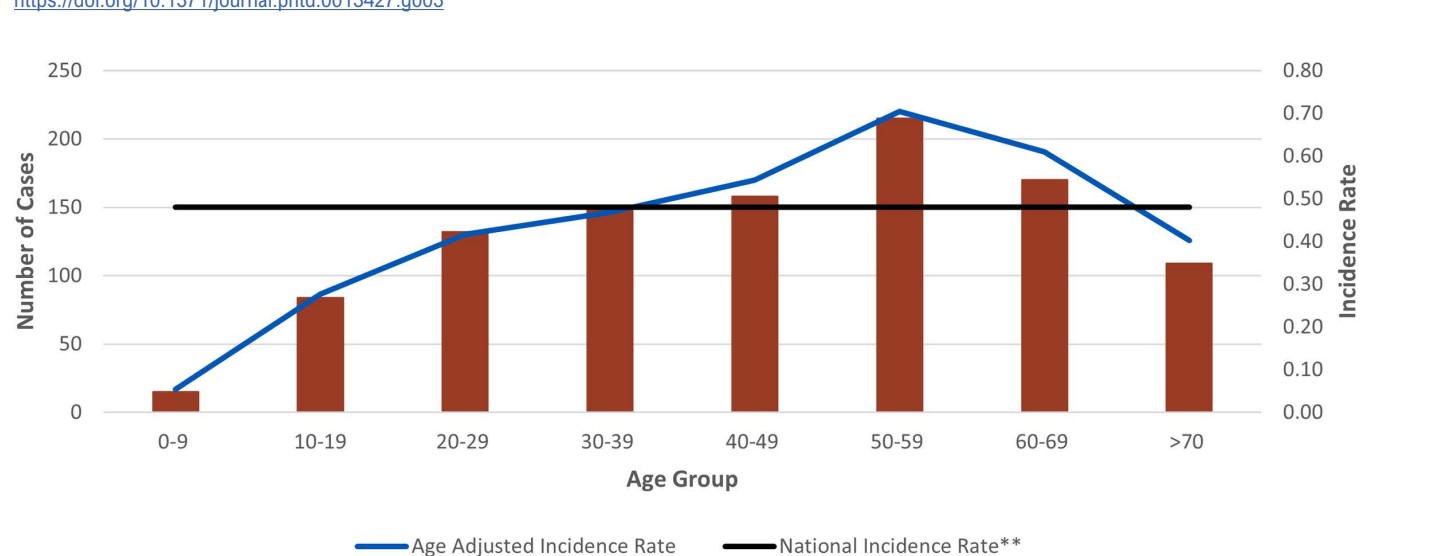

**Fig 4. Number of leptospirosis cases and incidence rate by 10-year age group - National Notifiable Diseases Surveillance System, United States, 2014-2020\*.** *Age unknown for 21 cases. ** National incidence rate includes jurisdictions that reported ≥1 case.

CSTE case classification criteria (Table 3). More probable cases met the supportive laboratory findings and one severe clinical manifestation (aseptic meningitis, gastrointestinal symptoms, pulmonary complications, cardiac arrhythmias, renal insufficiency, hemorrhage or jaundice with acute renal failure, n = 155/328, [47%]) than the supportive laboratory findings and two mild/non-specific symptoms (myalgia, headache, jaundice, conjunctival suffusion, rash, n = 138/328, [43%]). Less than 1% of probable cases (n = 2/328, [0.6%]) met the clinical criteria and reported involvement in an exposure event (adventure race, triathlon, flooding). Probable cases more frequently met the lower MAT titer criteria (≥200 but <800,

**Table 2. Number and percentage of leptospirosis cases (n=809) by occupation, exposures, antibiotic treatment received, and outcome – supplemental leptospirosis case report form, United States, 2014-2020.**

| | Confirmed Cases n=481 No. (%) | Probable Cases n=328 No. (%) | Total Cases n=809 No. (%) |
|---|---|---|---|
| **Occupation*** | **192 (40)** | **104 (32)** | **292 (36)** |
| *Animal Occupations* | *43 (22)* | *24 (23)* | *67 (23)* |
| • Rancher or farmer (animals) | 33 (17) | 21 (20) | 54 (18) |
| • Veterinary staff (e.g., vet, vet tech, or vet assistant) | 3 (1.6) | 1 (1.0) | 4 (1.4) |
| • Pet care (e.g., boarding, grooming) | 4 (2.1) | 0 (0) | 4 (1.4) |
| • Butcher or slaughterhouse worker | 2 (1.0) | 1 (1.0) | 3 (1.0) |
| • Pest or animal control | 1 (0.5) | 1 (1.0) | 2 (0.7) |
| *Environmental Occupations* | *97 (51)* | *51 (49)* | *148 (51)* |
| • Farmer (crops) | 51 (27) | 32 (31) | 79 (27) |
| • Construction and maintenance | 28 (15) | 8 (7.7) | 33 (11) |
| • Landscaping or yard care | 11 (5.7) | 8 (7.7) | 17 (5.8) |
| • Sanitation worker | 4 (2.1) | 2 (1.9) | 6 (2.1) |
| • Freshwater fisherman | 2 (1.0) | 1 (1.0) | 3 (1.0) |
| • Sewer or septic tank worker | 1 (0.5) | 0 (0) | 1 (0.3) |
| *Emergency Response Occupations* | 6 (3.1) | 7 (6.7) | 13 (4.5) |
| • Other frontline worker | 3 (1.6) | 3 (2.9) | 6 (2.1) |
| • Military | 1 (0.5) | 2 (1.9) | 3 (1.0) |
| • First responder | 2 (1.0) | 0 (0) | 2 (0.7) |
| • Postal or shipping worker | 0 (0) | 2 (1.9) | 2 (0.7) |
| Other | 27 (14) | 10 (10) | 37 (13) |
| Unknown | 29 (10) | 12 (12) | 31 (11) |
| **Reported Animal Contact** | **374 (78)** | **249 (76)** | **623 (77)** |
| Yes | 228 (61) | 156 (63) | 384 (62) |
| No | 81 (22) | 42 (17) | 123 (20) |
| Unknown | 65 (17) | 51 (20) | 116 (18) |
| **Specific Animal Contact*** | | | |
| Rats or mice | 99 (26) | 59 (24) | 158 (25) |
| Dog | 104 (28) | 79 (32) | 183 (29) |
| Cow | 11 (3.1) | 15 (6.1) | 26 (4.3) |
| Horse | 27 (7.2) | 19 (7.6) | 46 (7.4) |
| Domestic pig | 16 (4.3) | 8 (3.2) | 24 (3.9) |
| Goat or sheep | 9 (2.4) | 9 (3.6) | 18 (2.9) |
| Unknown animal | 2 (0.5) | 2 (0.8) | 4 (0.6) |
| Other animal | 76 (20) | 53 (21) | 125 (21) |
| • Chicken(s) | 34 (9.1) | 9 (3.6) | 43 (6.9) |
| • Cat(s) | 29 (7.8) | 20 (8.0) | 49 (7.9) |
| • Livestock | 9 (2.4) | 6 (2.4) | 15 (2.4) |
| • Rabbits | 6 (1.6) | 4 (1.6) | 10 (1.6) |
| **Reported Fresh Water or Mud Contact** | **339 (70)** | **239 (73)** | **578 (71)** |
| Yes | 163 (48) | 105 (44) | 268 (46) |
| No | 112 (33) | 79 (33) | 191 (33) |
| Unknown | 64 (19) | 55 (23) | 119 (21) |

*(Continued)*

| | Confirmed Cases<br>n=481<br>No. (%) | Probable Cases<br>n=328<br>No. (%) | Total Cases<br>n=809<br>No. (%) |
|---|---|---|---|
| **Specific Fresh Water or Mud Contact*** | | | |
| River/stream (running water) | 81 (24) | 57 (24) | 138 (24) |
| Lake/pond (still water) | 53 (16) | 38 (16) | 91 (16) |
| Flood water | 24 (7.1) | 11 (4.6) | 35 (6.1) |
| Rainwater run-off/puddles | 12 (3.5) | 2 (0.8) | 14 (2.4) |
| Mud | 53 (16) | 38 (16) | 91 (16) |
| Other fresh water or mud[1] | 21 (6.2) | 20 (8.4) | 41 (7.3) |
| Unknown fresh water or mud | 1 (0.3) | 0 (0) | 1 (0.2) |
| **Reported Sewage Contact** | 49 (10) | 17 (5.2) | 66 (8.2) |
| Yes | 17 (35) | 5 (29) | 22 (33) |
| No | 18 (37) | 8 (47) | 26 (40) |
| Unknown | 14 (28) | 4 (24) | 18 (27) |
| **Total Number of Animal and/or Environmental Contact Types[2]** | | | |
| Mean (range) | 2.1 (1–12) | 2.1 (1–7) | 2.1 (1–12) |
| 1 | 148 (49) | 93 (48) | 241 (49) |
| 2 | 71 (24) | 37 (19) | 108 (22) |
| 3 | 38 (13) | 27 (14) | 65 (13) |
| 4 | 24 (7.9) | 19 (10) | 43 (8.7) |
| 5 or more | 21 (7.0) | 16 (8.3) | 37 (7.5) |
| **Activity that Led to Animal or Environmental Contact*** | **475 (99)** | **326 (99)** | **801 (99)** |
| Avocational | 159 (33) | 91 (28) | 250 (31) |
| Occupational | 100 (21) | 63 (19) | 163 (20) |
| Recreational | 105 (22) | 98 (30) | 203 (25) |
| Other | 31 (6.5) | 15 (4.6) | 46 (5.7) |
| Unknown | 81 (17) | 58 (18) | 139 (17) |
| **Response to Exposure to Significant Weather**** | 76 (16) | 46 (14) | 122 (15) |
| Yes | 68 (89.5) | 42 (91.3) | 110 (90.2) |
| No | 2 (2.6) | 3 (6.5) | 5 (4.1) |
| Unknown | 6 (7.9) | 1 (2.2) | 7 (5.7) |
| **Response to Travel** | **371 (77)** | **233 (71)** | **604 (75)** |
| Yes | 66 (18) | 72 (31) | 138 (23) |
| No | 261 (70) | 134 (57) | 395 (65) |
| Unknown | 44 (12) | 27 (12) | 71 (12) |
| **Travel Location** | | | |
| Travelled domestically (≥50 miles from residence) | 27 (7.3) | 26 (11) | 53 (8.8) |
| Travelled internationally | 31 (8.4) | 39 (17) | 70 (12) |
| Travelled domestically & internationally | 5 (1.3) | 6 (2.6) | 11 (1.8) |
| **Response to Antibiotic Treatment Prescribed or Administered for Illness** | **339 (70)** | **228 (70)** | **567 (70)** |
| Yes | 310 (91) | 197 (86) | 507 (89) |
| No | 8 (2.4) | 15 (6.6) | 23 (4.1) |
| Unknown | 21 (6.2) | 16 (7.0) | 37 (6.5) |
| **Specific Antibiotic Prescribed or Administered for Illness[3]*** | | | |
| Amoxicillin | 8 (2.6) | 7 (3.6) | 15 (3.0) |

*(Continued)*

**Table 2.** (Continued)

| | Confirmed Cases n=481 No. (%) | Probable Cases n=328 No. (%) | Total Cases n=809 No. (%) |
|---|---|---|---|
| Ampicillin | 3 (1.0) | 3 (1.5) | 6 (1.2) |
| Azithromycin | 16 (5.2) | 7 (3.6) | 23 (4.5) |
| Ceftriaxone | 119 (38) | 61 (31) | 180 (36) |
| Ciprofloxacin | 6 (1.9) | 3 (1.5) | 9 (1.8) |
| Doxycycline | 164 (53) | 94 (48) | 258 (51) |
| Levofloxacin | 41 (4.5) | 5 (2.5) | 19 (3.7) |
| Penicillin | 29 (9.4) | 18 (9.1) | 47 (9.3) |
| Vancomycin | 35 (11) | 17 (8.6) | 52 (10) |
| Antibiotic Unknown | 66 (21) | 41 (21) | 107 (21) |
| **Hospitalized** | **425 (88)** | **284 (87)** | **709 (88)** |
| Yes | 371 (87) | 235 (83) | 606 (85) |
| No | 52 (12) | 47 (16) | 99 (14) |
| Unknown | 2 (0.5) | 2 (0.7) | 4 (0.6) |
| **Outcome** | **424 (88)** | **288 (88)** | **712 (88)** |
| Recovered | 309 (73) | 199 (69) | 508 (71) |
| Died | 40 (9.4) | 34 (12) | 74 (10) |
| Other (still sick, still hospitalized) | 15 (3.5) | 12 (4.2) | 27 (3.8) |
| Unknown | 60 (14) | 43 (15) | 103 (14) |

*Responses not mutually exclusive.

**Significant weather included flooding, heavy rainfall, hurricane, cyclone, typhoon, windstorm, tornado, earthquake, or mudslide near an individual's home, work site, recreational activities, or travel location.

1 Ocean, cistern, pool, well water.

2 Number of exposures were calculated for cases that reported contact with animals, freshwater, mud, or sewage and at least one type of animal or environmental (fresh water, mud, sewage) contact.

3 Timing of antibiotics is not available; unknown if multiple antibiotics were taken concurrently.

n=112/328, [34%])) or were IgM ELISA positive (n=141/328, [43%]), two of the four supportive laboratory criteria. Confirmed cases were identified by detection of pathogenic *Leptospira* DNA by PCR (n=210/481, [44%]), the higher MAT titer criteria (≥800), or four-fold rise between acute and convalescent serum samples (n=199/481, [41%]).

Data completeness on the supplemental CRF varied by section; activities that led to exposure(s) (n=801/809, [99%]) and signs and symptoms (n=778/809, [96%]) data were the most complete. Fever (n=628/778, [81%]), myalgia (n=489/778, [63%]), headache (n=422/778, [54%]), vomiting and nausea (n=277/778, [36%]), jaundice (n=226/778, [29%]), and acute renal insufficiency (n=220/778, [28%]) were the most frequently reported signs and symptoms (Fig 5). For cases with illness onset date and specimen collection date (n=438/809, [54%]), a median of 5 days (range: 41 days) elapsed between onset of symptoms and collection of a specimen for leptospirosis testing.

Antibiotics were prescribed or administered for 89% (n=507/567) of the cases with antibiotic data submitted. For cases with illness onset date and antibiotic start date (n=218/809, [27%]), a median of 5 days (range: 32 days) elapsed between illness onset date and the start of antibiotic treatment. Doxycycline (n=258/507, [51%]) and ceftriaxone (n=180/507, [36%]) were the most frequently prescribed or administered antibiotics. 85% (n=606/709) of cases with data were hospitalized. For cases with illness onset date and hospital admission date (n=537, [66%]), a median of 4 days elapsed between onset of symptoms and hospitalization (range: 85 days). For cases with length of hospitalization (n=284/809, [35%]), the median number of days hospitalized was 12 (range: 81 days).

**Table 3. Number and percentage of reported and confirmed cases that met the Council for State and Territorial Epidemiologists (CSTE) case definition by clinical and laboratory criteria – supplemental case report form, United States, 2014-2020.**

| | Confirmed Cases n = 481 No. (%) | Probable Cases n = 328 No. (%) |
|---|---|---|
| **Met CSTE case definition** | 461 (96) | 187 (57) |
| **Clinical criteria** | | |
| No clinical criteria reported | 25 (5.2) | 5 (1.5) |
| Met fever & 1 severe clinical manifestation* | 113 (23) | 57 (17) |
| Met fever & 2 mild/non-specific symptoms** | 40 (8.3) | 37 (11) |
| Met both clinical criteria | 177 (37) | 128 (39) |
| **Exposure event** | | |
| Reported an exposure event | 2 (0.4) | 0 (0) |
| **Laboratory results** | | |
| No laboratory testing reported | 15 (3.1) | 32 (10) |
| *Supportive laboratory criteria* | | |
| IgM+ | 129 (27) | 141 (43) |
| MAT+*** | 18 (3.7) | 112 (34) |
| IHC+ | 5 (1.0) | 0 (0) |
| IgM+ & MAT+*** | 14 (2.9) | 32 (10) |
| IgM+ & IHC+ | 1 (0.2) | 0 (0) |
| *Confirmatory laboratory criteria* | | |
| PCR+ | 210 (44) | 1 (0.3) |
| MAT+**** | 172 (36) | 0 (0) |
| MAT 4-fold rise**** | 2 (0.4) | 0 (0) |
| Culture+ | 19 (4.0) | 0 (0) |
| MAT+**** & MAT 4-fold rise**** | 25 (5.2) | 0 (0) |
| PCR+ & Culture+ | 2 (0.4) | 0 (0) |
| PCR+ & MAT+**** | 25 (5.2) | 0 (0) |
| MAT+**** & Culture+ | 2 (0.4) | 0 (0) |
| PCR+, MAT+****, MAT 4-fold rise**** | 3 (0.6) | 0 (0) |
| PCR+, Culture+, MAT+**** | 1 (0.2) | 0 (0) |

*Severe clinical manifestation include aseptic meningitis, gastrointestinal symptoms (e.g., abdominal pain, nausea, vomiting, diarrhea), pulmonary complications (e.g., cough, breathlessness, hemoptysis), cardiac arrhythmias or electrocardiogram abnormalities, renal insufficiency (e.g., anuria, oliguria), hemorrhage (e.g., intestinal, pulmonary, hematuria, hematemesis), or jaundice with acute renal failure.

**Mild/non-specific symptoms include myalgia, headache, jaundice, conjunctival suffusion with purulent discharge, or rash (i.e., maculopapular or petechial).

***Supportive laboratory criteria threshold for MAT positive titer (probable case) is 1: ≥ 200 but 1: < 800 in ≥1 serum specimens.

****Confirmatory laboratory criteria threshold for a MAT positive titer (confirmed case) is 1: ≥ 800 ≥1 serum specimens or ≥4-fold rise in titer between acute and convalescent-phase serum specimens.

The distribution of cases by sex and age was similar between NNDSS data and supplemental case data (83% [n = 668] of cases were male and 17% [n = 139] were female in the supplemental case data; sex was unknown for two cases; median age was 47 years in the supplemental case data). For cases with both hospitalization and sex data (n = 606/809, [75%]), 86% of cases among males were hospitalized compared to 82% of cases among females. Individuals 60+ years had the highest hospitalization percentage (93%), followed by 50–59 years (88%), 30–39 years (87%), and 40–49 years (86%). Of the 712 cases that reported patient outcome, 10% (n = 74/712) died and 71% (n = 508/712) recovered. Among

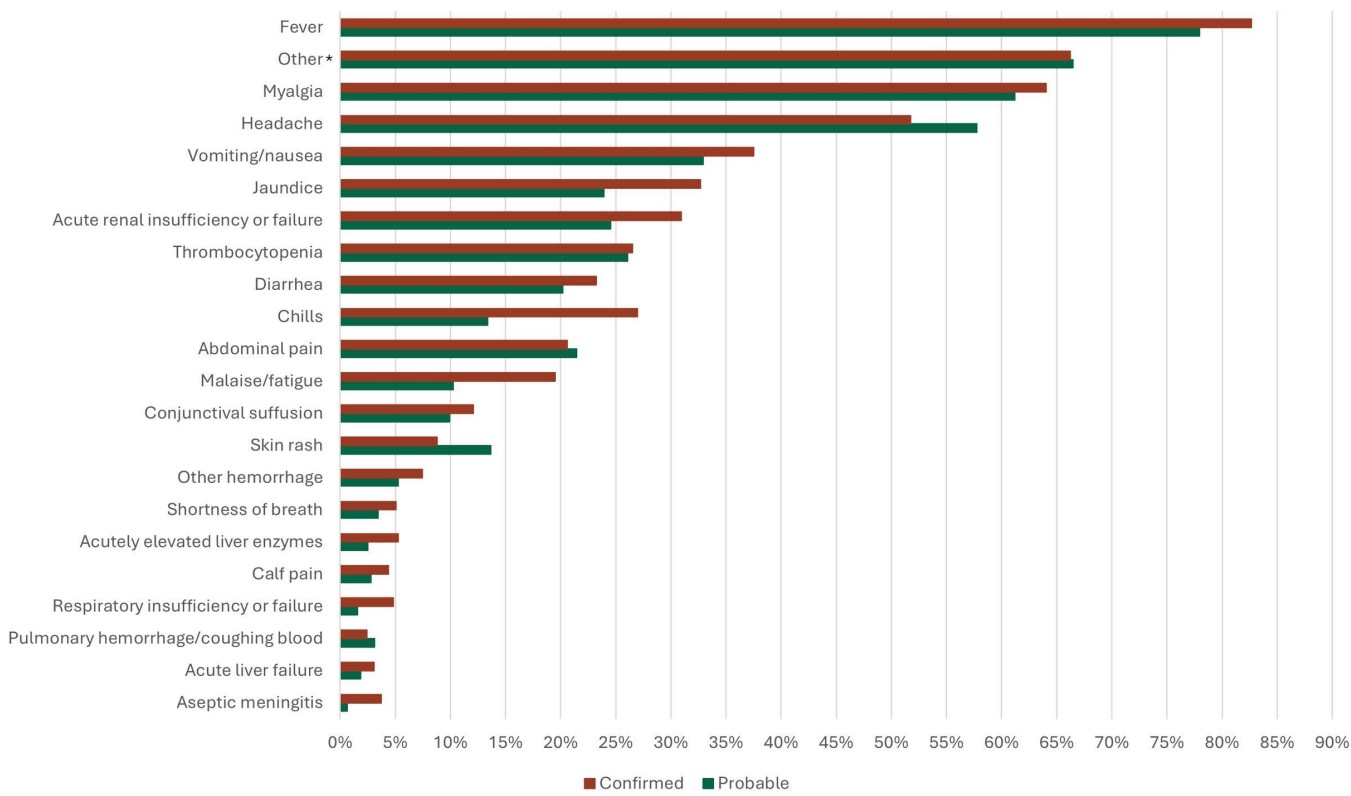

**Fig 5. Frequency of reported leptospirosis signs and symptoms – supplemental case report form, United States, 2014-2020.** *Other symptoms included anorexia, abnormal pain, cardiac involvement, cough, dizziness, gastrointestinal involvement, low blood pressure, photophobia, pulmonary complications, rhinitis, sore throat, weakness, weight loss.

those that died, 55% of cases were in individuals aged 50+years. Among fatal cases, median time between illness onset and hospitalization was 5 days (range: 31 days); median time between illness onset and start of antibiotic treatment was 9 days (range: 12 days). There was no difference in time between illness onset and collection of a specimen for leptospirosis testing between fatal and non-fatal cases.

Crop farmers (n=79/292, [27%]), animal farmers (n=54/292, [18%]), and construction or maintenance workers (n=33/292, [11%]) accounted for half of leptospirosis cases with occupation data (n=292/809, [36%]). Data on animal and freshwater or mud exposures was completed for 77% (n=623/809) and 71% (n=578/809) of supplemental CRFs, respectively. Most cases reported contact with animals or their bodily fluids (n=384/623, [62%]). Contact with dogs (n=183/623, [29%]), rats or mice (n=158/623, [25%]), and horses (n=46/623, [7.4%]) were the most frequently reported animal contact. Nearly half of cases (n=268/578, [46%]) reported contact with freshwater or mud; running water (e.g., river/stream, n=138/578, [24%]), mud (n=91/578, [16%]), and still water (e.g., lake/pond, n=91/578, [16%]) were the most frequently reported exposures. For cases with sewage contact data (n=66/809, [8.2%]), 33% (n=22/66) reported contact with sewage. Cases averaged 2.1 exposure types (distinct animal species, water, mud, or sewage exposures) with a range of 1–12 exposure types reported.

More cases reported avocational activities (n=250/801, [31%]) as the source of their animal or environmental exposure(s) than recreational (n=203/801, [25%]) or occupational (n=163/801, [20%]) activities. Pet or livestock ownership was the most frequent activity reported (n=173/801, [22%]). Recreational activities were the most common exposures among cases aged 0–39 years (0–9 years, n=8/15, [53%]; 10–19 years, n=39/77, [51%]; 20–29 years, n=38/112, [34%];

30–39 years, n = 33/113, [29%]) and avocational activities among cases aged 40–70 + years (40–49 years, n = 43/138, [31%]; 50–59 years, n = 44/149, [30%]; 60–69 years, n = 52/132, [39%]; 70 + years, n = 28/61, [46%]).

75% (n = 604) of cases reported whether or not they traveled (≥50 miles from normal residence) in the 30 days before illness onset. Among cases that reported travel (n = 138/604, [23%]), 12% (n = 70) reported international travel, 8.8% (n = 53) reported domestic travel ≥50 miles from residence, and 1.8% (n = 11) reported both international and domestic travel.

## Discussion

This report provides the first national summary of leptospirosis epidemiology in the U.S. since leptospirosis was reinstated as a nationally notifiable condition in 2014. Leptospirosis is an increasingly widespread cause of illness in the U.S.; cases of leptospirosis were reported from most states and jurisdictions (n = 35/62, [65%]) and caused substantial morbidity (85% of cases were hospitalized) and mortality (10% of cases died). The incidence of leptospirosis between 2014–2020 nearly tripled when compared to 1988–1994 (0.48 vs 0.17 per 100,000 persons) [22].

In this report, 85% of leptospirosis patients were hospitalized. Hospitalization is one of the most expensive types of healthcare use [30]. A previous study using population hospital discharge data for 1998–2009 from the Nationwide Inpatient Sample (NIS) concluded that the average length of stay for leptospirosis-associated hospitalizations was 6.9 days and cost an average of $39,181 [31] which would be $55,647.88 in 2023, when adjusted for inflation. This average cost is likely an underestimation as it only includes direct hospitalization medical costs and does not consider outpatient medical expenses, non-medical costs, and productivity losses. More cases met the CSTE clinical criteria that included fever and one severe clinical manifestation (e.g., pulmonary complications, hemorrhage, renal failure) than clinical criteria that included fever and two mild/non-specific symptoms (headache, myalgia, rash, etc). The frequency of more severe clinical manifestations and high percentage of hospitalization among reported cases may indicate under-identification of milder disease, healthcare seeking bias, or disease-severity bias in reporting. Cases with illness requiring hospitalization may undergo more testing to guide care and therefore be more often diagnosed with leptospirosis and reported to the CDC. Identification of leptospirosis in its early stages relies on clinical suspicion based on a patient's risk factors, exposure history, and presenting signs and symptoms. This highlights the importance of clinician awareness of the varied, but milder, clinical presentations and of conducting thorough patient history and testing for early case identification and treatment.

In addition to delayed case identification, existing literature suggests that delayed antimicrobial therapy [32–34], infection with *L. interrogans* serogroup Icterohemorrhagiae [32–34], high levels of leptospiremia [32,33], chronic hypertension [33,34], and chronic alcoholism [33,34] are factors associated with severe leptospirosis, including death. Although the majority of cases (89%) received antibiotic treatment, nearly 10% of cases died, a higher case fatality than the median mortality estimate for passive surveillance or when cases encompass both rural and urban locations [2]. Severe cases should be treated with high doses of intravenous penicillin or ceftriaxone [35,36]. On the other hand, mild illness can be treated with oral antibiotics such as doxycycline, azithromycin, ampicillin, amoxicillin, or third-generation cephalosporins. In our surveillance data, most cases received antibiotics recommended in current treatment guidelines [36]. Many cases received vancomycin which is not recommended for the treatment of leptospirosis [35]. This suggests that diagnosis and treatment specific for leptospirosis was delayed. Future work should prioritize identifying risk factors, predictors of severe leptospirosis, and provider education about appropriate antibiotic selection when leptospirosis is in the differential diagnosis to improve clinical management and outcomes.

Nonetheless, national surveillance data indicates significant under-recognition and under-reporting of leptospirosis cases to NNDSS. The high percentage of cases that were hospitalized likely indicates bias in case identification and reporting of severe disease. Due to its varied, but non-specific presentation, leptospirosis can be confused with other acute febrile illnesses [6,37] or individuals may experience a self-limiting disease, contributing to case under-recognition

[7]. Additionally, individuals with clinically compatible illness may not seek healthcare and access to healthcare may be disrupted during extreme weather events [38]. Furthermore, over the study time period, just 65% of jurisdictions reported a leptospirosis case to CDC due to the inability of jurisdictions to report cases to CDC where leptospirosis is not locally mandated and changes to reportability in many jurisdictions over some of this report's time period.

Despite the challenges of under-recognition and under-reporting of cases, several important epidemiological trends emerged underscoring known and changing risk factors for leptospirosis infection in the U.S. Four out of every five reported cases were males, with males aged 40–69 years experiencing the highest age-adjusted incidence rates. Multiple other studies have reported similar findings [39–43], a higher proportion of cases among males, thought to be a result of exposure-related bias. Occupation associated contact with infected animals and contaminated water or mud are well-documented risk factors for leptospirosis [23,24,44,45]. However, in our analysis, we found more cases reported avocational exposures like pet or livestock ownership or gardening and yard work than occupational exposures. This trend held regardless of the case's reported sex. Large outbreaks of leptospirosis have been reported as a result of avocational and recreational exposures [10,14,46]. Although cases frequently reported pet or livestock ownership, several studies have found that despite high-risk exposures, there was little to no evidence of zoonotic transmission during outbreaks of leptospirosis in dogs [47,48]. However, the changing epidemiological trend from occupational to avocational and recreational exposures highlights the need for interventions that mitigate transmission in these environments.

In our surveillance data, the majority of cases acquired leptospirosis domestically. Average weekly case counts rose in July and peaked in October. July-October is summer and early fall in the U.S. when temperatures are higher [49]. Many Americans participate in outdoor activities in the summer and fall [50], when environmental conditions are more suitable for the persistence of *Leptospira* bacteria in water and soil [51]. Hurricane season occurs in the Atlantic between June 1-November 30 and in the Pacific between May 15- November 30 [52]. In September 2017, the year with the highest number of annual cases, Hurricane Irma struck the U.S. Virgin Islands and Hurricane Maria struck Puerto Rico and the U.S. Virgin Islands. Puerto Rico reported an increase in cases of leptospirosis [53], and the U.S. Virgin Islands reported their first human cases of leptospirosis after the hurricanes [54]. Leptospirosis cases are expected to increase as climate change causes higher temperatures and more frequent, intense extreme weather events, such as hurricanes and floods [13]. Climate projections for the U.S. include higher temperatures and more frequent, heavy precipitation [55]. Heavy rainfall and flooding increase the risk of leptospirosis by bringing bacteria and their animal hosts into closer contact with humans and increasing environmental exposures through contact with contaminated freshwater and wet soil [13]. Furthermore, leptospires are also able to survive for longer periods of time in higher temperatures and humid environments [19]. Global warming may lengthen seasons and expand the geographical areas for optimal survival and transmission of leptospires [13,56]. Higher temperatures can also encourage water-based activities for both humans and animals, promoting more contact between humans, livestock, pets, wildlife, and potentially contaminated water sources [57]. In urban areas, exposure to rodents and the proliferation of leptospires in warmer and wetter weather has led to an increase in cases [58]. Markedly higher incidence rates in Puerto Rico and Hawaii warrant further investigation into risk factors, exposures, provider education, and public information approaches for disease control in these tropical environments. Information from these jurisdictions may be informative to national disease control efforts as climate change increasingly creates conditions for leptospirosis spread in other regions.

## Limitations

The findings in this report are subject to at least five limitations. As discussed, leptospirosis case counts are likely underestimating the actual case counts because of under-diagnosis, under-reporting, and under-ascertainment. Second, disease severity and mortality are likely overestimated due to bias toward case ascertainment among individuals experiencing more severe disease. Third, despite the potential for increased incidence during large scale flooding, cyclone, or hurricanes events, disruptions to medical care and diagnostic infrastructure because of these events may lead to

under-reporting. Fourth, multiple variables (race, ethnicity, occupation) had incomplete data. Jurisdictions may collect different data as part of case investigation, the patient may be lost to follow-up, or the patient may not disclose the information [59,60]. Missing and unknown data on race, ethnicity, and occupation limit the ability to identify and estimate disparities, make recommendations to improve early diagnosis and treatment, and develop prevention recommendations. Finally, responses to animal and environmental exposures, travel history, and clinical information and presentation are subject to recall bias.

## Conclusions

An increasing number of individuals in the U.S. are being diagnosed with leptospirosis, mostly from domestic sources of infection. Given under-reporting and bias toward severe disease reporting, increased case ascertainment and reporting consistency are needed to better understand the clinical and epidemiological patterns of this disease. Continued surveillance, including enhanced surveillance in jurisdictions with known high incidence, adding surveillance from jurisdictions where leptospirosis may not yet be reportable, and streamlining supplemental case data reporting by moving away from fax and email, would allow public health professionals to better understand trends and changing epidemiology, identify populations at risk, and more effectively guide public health interventions. This improved understanding will inform disease control efforts including provider education and public information for disease prevention, detection, and treatment. These measures are particularly urgent for leptospirosis, given the anticipated impacts of climate change on disease distribution and incidence. The rapid increase in incidence, markedly higher incidence in tropical jurisdictions, and anticipated warmer temperatures and more frequent, heavy rainfall in the coming years for parts of the continental U.S indicate the leptospirosis will occur more frequently and with more widespread impact.

## Acknowledgments

State and territorial health departments that voluntarily reported leptospirosis case data; Ian Dunn from the CDC's Geospatial Research, Analysis and Services Program (GRASP); John Morris, Nay Newman, Benjamin Schneider, Jill Perry, Kaitlin Polly, Zahiduzzaman Biswas, Jonathan Batross, Kayla Janos, Mark Borys, and PJ Jarquin from the CDC's DCI-PHER team

The findings and conclusions in this report are those of the authors and do not necessarily represent the official position of the U. S. Centers for Disease Control and Prevention.

## Author contributions

**Conceptualization:** Christine Atherstone, Renee Galloway, Ilana Schafer, William A. Bower, Maria E. Negron, Katherine DeBord.

**Data curation:** Christine Atherstone, Renee Galloway, Ilana Schafer, Aileen Artus, Melissa Marzan Rodriguez, Kyle Ryff, Abigail Medina Rivera, Sally Slavinski, Marc Paladini, Sarah K. Kemble, Janet M. Berreman, Grayson Kallas, Rita Traxler, Grishma Kharod, Marta Guerra, Robyn A. Stoddard, Hannah Moore, Katherine DeBord.

**Formal analysis:** Christine Atherstone, Renee Galloway, Ilana Schafer, Aileen Artus, Rita Traxler, Grishma Kharod, Marta Guerra, Hannah Moore, Katherine DeBord.

**Investigation:** Christine Atherstone, Renee Galloway, Ilana Schafer, Aileen Artus, Melissa Marzan Rodriguez, Kyle Ryff, Abigail Medina Rivera, Sally Slavinski, Marc Paladini, Sarah K. Kemble, Janet M. Berreman, Grayson Kallas, Rita Traxler, Grishma Kharod, Marta Guerra, Robyn A. Stoddard, Katherine DeBord.

**Methodology:** Christine Atherstone, Renee Galloway, Ilana Schafer, Aileen Artus, Rita Traxler, Grishma Kharod, Marta Guerra, Maria E. Negron, Katherine DeBord.

**Project administration:** Christine Atherstone, Renee Galloway, Ilana Schafer, Aileen Artus, Melissa Marzan Rodriguez, Kyle Ryff, Abigail Medina Rivera, Sally Slavinski, Marc Paladini, Sarah K. Kemble, Janet M. Berreman, Grayson Kallas, Rita Traxler, Grishma Kharod, Marta Guerra, Robyn A. Stoddard, William A. Bower, Maria E. Negron, Katherine DeBord.

**Resources:** Christine Atherstone, Renee Galloway, Melissa Marzan Rodriguez, Kyle Ryff, Abigail Medina Rivera, Sally Slavinski, Marc Paladini, Sarah K. Kemble, Janet M. Berreman, Grayson Kallas, Rita Traxler, Robyn A. Stoddard, Katherine DeBord.

**Software:** Christine Atherstone, Hannah Moore, Katherine DeBord.

**Supervision:** Christine Atherstone, Renee Galloway, Melissa Marzan Rodriguez, Kyle Ryff, Sally Slavinski, Sarah K. Kemble, Janet M. Berreman, William A. Bower, Maria E. Negron.

**Validation:** Christine Atherstone, Renee Galloway, Ilana Schafer, Aileen Artus, Melissa Marzan Rodriguez, Kyle Ryff, Abigail Medina Rivera, Sally Slavinski, Marc Paladini, Sarah K. Kemble, Janet M. Berreman, Grishma Kharod, Marta Guerra, Hannah Moore, Katherine DeBord.

**Visualization:** Christine Atherstone.

**Writing – original draft:** Christine Atherstone, Maria E. Negron, Katherine DeBord.

**Writing – review & editing:** Christine Atherstone, Renee Galloway, Ilana Schafer, Aileen Artus, Melissa Marzan Rodriguez, Kyle Ryff, Abigail Medina Rivera, Sally Slavinski, Marc Paladini, Sarah K. Kemble, Janet M. Berreman, Grayson Kallas, Rita Traxler, Grishma Kharod, Robyn A. Stoddard, Hannah Moore, William A. Bower, Maria E. Negron, Katherine DeBord.

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
