## [Decision Letter · Decision Letter 0]

22 May 2025

Response to Reviewers
Revised Manuscript with Track Changes
Manuscript

Shaden Kamhawi

co-Editor-in-Chief

Paul Brindley

co-Editor-in-Chief

**Journal Requirements:**

At this stage, the following Authors/Authors require contributions: Christine Atherstone, Renee Galloway, Ilana Schafer, Aileen Artus, Melissa Marzan Rodriguez, Kyle Ryff, Abigail Medina Rivera, Sally Slavinski, Marc Paladini, Sarah K. Kemble, Janet M. Berreman, Grayson Kallas, Rita Traxler, Grishma Kharod, Marta Guerra, Robyn A. Stoddard, Hannah Moore, William A. Bower, Maria E. Negron, and Katherine DeBord. Please ensure that the full contributions of each author are acknowledged in the "Add/Edit/Remove Authors" section of our submission form.

- ® on page: 20.

Potential Copyright Issues:

- Figure 1. Please (a) provide a direct link to the base layer of the map (i.e., the country or region border shape) and ensure this is also included in the figure legend; and (b) provide a link to the terms of use / license information for the base layer image or shapefile. We cannot publish proprietary or copyrighted maps (e.g. Google Maps, Mapquest) and the terms of use for your map base layer must be compatible with our CC BY 4.0 license.

**Reviewers' comments:**

**Key Review Criteria Required for Acceptance?**

**Methods**

-Are the objectives of the study clearly articulated with a clear testable hypothesis stated?

-Is the study design appropriate to address the stated objectives?

-Is the population clearly described and appropriate for the hypothesis being tested?

-Is the sample size sufficient to ensure adequate power to address the hypothesis being tested?

-Were correct statistical analysis used to support conclusions?

-Are there concerns about ethical or regulatory requirements being met?

Reviewer #1: a minor question: why stopping at 2020?

**Results**

-Does the analysis presented match the analysis plan?

-Are the results clearly and completely presented?

-Are the figures (Tables, Images) of sufficient quality for clarity?

Reviewer #1: A major issue here is how can one correlate human leptospirosis to pet ownership per se. Many people have pets, dogs for example. To persuasively attribute human disease to dog (for example) ownership, you have to either demonstrate animal infection or 2. demosntrate increased prevalence of human disease to matched controls with no dogs.

**Conclusions**

-Are the conclusions supported by the data presented?

-Are the limitations of analysis clearly described?

-Do the authors discuss how these data can be helpful to advance our understanding of the topic under study?

-Is public health relevance addressed?

Reviewer #1: 1. Based on my reservation mentioned above, I am not certain that a switch towards a predominantly avocational correlation of human disease can be strongly supported.

2. the minor possibility that increased incidence in recent years may be due to increased awareness and thus testing, or improved diagnostic techniques, should be addressed (although I do not consider it the reason of the tripled incidence)

**Editorial and Data Presentation Modifications?**

Reviewer #1: 1. a minor error: in Surveillance Case Definitions the authors mention that the disease was reinstated as notifiable in 2013. While previously the year of reinstating is mentioned as being 2014.

2. the authors erroneously refer to ceftriaxone and cefotaxime as orally administered antibiotics.

**Summary and General Comments**

Reviewer #1: My major concern is the correlation of disease to pet ownership without further evidence. All other are minor, but this changes the conclusions of the manuscript.

PLOS authors have the option to publish the peer review history of their article (what does this mean? ). If published, this will include your full peer review and any attached files.

**Do you want your identity to be public for this peer review?** For information about this choice, including consent withdrawal, please see our Privacy Policy .

Reviewer #1: **Yes: ** Georgios Pappas

**Figure resubmission:****Reproducibility:** To enhance the reproducibility of your results, we recommend that authors of applicable studies deposit laboratory protocols in protocols.io, where a protocol can be assigned its own identifier (DOI) such that it can be cited independently in the future. Additionally, PLOS ONE offers an option to publish peer-reviewed clinical study protocols. Read more information on sharing protocols at https://plos.org/protocols?utm_medium=editorial-email&utm_source=authorletters&utm_campaign=protocols

---

## [Editor Report · Decision Letter 1]

31 Jul 2025

Dear Dr Atherstone,

We are pleased to inform you that your manuscript 'Epidemiological, Temporal, and Geographic Trends of Leptospirosis in the United States, 2014-2020' has been provisionally accepted for publication in PLOS Neglected Tropical Diseases.

Best regards,

Joseph M. Vinetz

Section Editor

Joseph Vinetz

Section Editor

Shaden Kamhawi

co-Editor-in-Chief

Paul Brindley

co-Editor-in-Chief

---

## [Editor Report · Acceptance letter]

Dear Dr Atherstone,

We are delighted to inform you that your manuscript, " 

Epidemiological, Temporal, and Geographic Trends of Leptospirosis in the United States, 2014-2020," has been formally accepted for publication in PLOS Neglected Tropical Diseases.

Best regards,

Shaden Kamhawi

co-Editor-in-Chief

Paul Brindley

co-Editor-in-Chief
